# Three-Dimensional Histological Characterization of the Placental Vasculature Using Light Sheet Microscopy

**DOI:** 10.3390/biom13061009

**Published:** 2023-06-17

**Authors:** Lennart Freise, Rose Yinghan Behncke, Hanna Helene Allerkamp, Tim Henrik Sandermann, Ngoc Hai Chu, Eva Maria Funk, Lukas Jonathan Hondrich, Alina Riedel, Christian Witzel, Nils Rouven Hansmeier, Magdalena Danyel, Alexandra Gellhaus, Ralf Dechend, René Hägerling

**Affiliations:** 1Research Group ‘Lymphovascular Medicine and Translational 3D-Histopathology’, Institute of Medical and Human Genetics, Charité - Universitätsmedizin Berlin, Corporate Member of Freie Universität Berlin and Humboldt-Universität zu Berlin, Augustenburger Platz 1, 13353 Berlin, Germany; lennart.freise@charite.de (L.F.); rose.behncke@charite.de (R.Y.B.); tim-henrik.sandermann@charite.de (T.H.S.); ngoc-hai.chu@charite.de (N.H.C.); eva-maria.funk@charite.de (E.M.F.); lukas-jonathan.hondrich@charite.de (L.J.H.); nils.hansmeier@charite.de (N.R.H.); magdalena.danyel@charite.de (M.D.); 2Berlin Institute of Health, Charité - Universitätsmedizin Berlin, BIH Center for Regenerative Therapies, Augustenburger Platz 1, 13353 Berlin, Germany; 3Department of Obstetrics and Gynecology, Medical University of Graz, Auenbruggerplatz 14, 8036 Graz, Austria; hanna.allerkamp@medunigraz.at; 4Department of Gynecology and Obstetrics, University Hospital Essen, 45147 Essen, Germany; alina.riedel@uk-essen.de (A.R.); alexandra.gellhaus@uk-essen.de (A.G.); 5Department of Surgery, Campus Charité Mitte and Campus Virchow-Klinikum, Charité - Universitätsmedizin Berlin, Corporate Member of Freie Universität Berlin and Humboldt-Universität zu Berlin, 10117 Berlin, Germany; christian.witzel@charite.de; 6Research Group ‘Development and Disease’, Max Planck Institute for Molecular Genetics, Ihnestraße 63–73, 14195 Berlin, Germany; 7Berlin Institute of Health, Charité - Universitätsmedizin Berlin, BIH Academy, Clinician Scientist Program, Charitéplatz 1, 10117 Berlin, Germany; 8Experimental and Clinical Research Center (ECRC), a Cooperation of Charité - Universitätsmedizin Berlin and Max Delbruck Center for Molecular Medicine (MDC), Lindenbergerweg 80, 13125 Berlin, Germany; ralf.dechend@charite.de; 9HELIOS Klinikum, 13125 Berlin, Germany

**Keywords:** light sheet microscopy, placenta, histopathology, lymphatic mimicry, pre-eclampsia

## Abstract

The placenta is the first embryonic organ, representing the connection between the embryo and the mother, and is therefore necessary for the embryo’s growth and survival. To meet the ever-growing need for nutrient and gas exchange, the maternal spiral arteries undergo extensive remodeling, thus increasing the uteroplacental blood flow by 16-fold. However, the insufficient remodeling of the spiral arteries can lead to severe pregnancy-associated disorders, including but not limited to pre-eclampsia. Insufficient endovascular trophoblast invasion plays a key role in the manifestation of pre-eclampsia; however, the underlying processes are complex and still unknown. Classical histopathology is based on two-dimensional section microscopy, which lacks a volumetric representation of the vascular remodeling process. To further characterize the uteroplacental vascularization, a detailed, non-destructive, and subcellular visualization is beneficial. In this study, we use light sheet microscopy for optical sectioning, thus establishing a method to obtain a three-dimensional visualization of the vascular system in the placenta. By introducing a volumetric visualization method of the placenta, we could establish a powerful tool to deeply investigate the heterogeneity of the spiral arteries during the remodeling process, evaluate the state-of-the-art treatment options, effects on vascularization, and, ultimately, reveal new insights into the underlying pathology of pre-eclampsia.

## 1. Introduction

The placenta is the first organ of every mammalian embryo. It represents the transient connection between fetus and mother during gestation and is mandatory for the exchange of gases, nutrients, and waste products [1]. Hence, the placenta plays a critical role during the most rapid growth of the evolving embryo. Insufficiencies or defects can lead to severe pregnancy-associated disorders, including but not limited to pre-eclampsia, fetal growth restriction, preterm birth, or even intrauterine lethality [2]. To face the ever-growing need for maternal resources for the fetus, the vasculature undergoes extensive remodeling. As a result, the spiral arteries loose smooth muscle cells and their autonomic innervation to enhance vasodilation [3]. In fact, arterial blood flow to the human placenta can increase from 45 mL/min before pregnancy to 750 mL/min during pregnancy [4].

### 1.1. The Anatomical Differences in the Placenta between Humans and Rodents Limit the Direct Reproducibility

This study aims to establish a new three-dimensional visualization technique of the placental vascular architecture and its remodeling, which can be applied to humans. Therefore, it is necessary to work with a model that combines an easy access, handling, and high reproducibility. Thus, laboratory rodents such as mice represent a convenient model for studying placentation [5]. Just like primates, mice and nearly all rodents possess a hemochorial type of placenta, an invasive form of placentation, marked by maternal blood directly bathing fetal trophoblast cells for maternal nutrient delivery [6]. However, in humans, the villi are encased by only one layer of syncytiotrophoblast cells (STBs), directly bathing the maternal blood, whereas the labyrinthine placenta in mice has two STB layers with the cytotrophoblast layer being in direct contact with maternal blood [7].

As the hemochorial placentation progresses, rodents such as rats and mice also share comparable characteristics in the architecture of the placenta and the organization of trophoblast cells within the placentation site. Generally, the placenta can be divided into a fetal and a maternal compartment [8,9]. Additionally, in all rodents and humans, the fetal region is delineated from the maternal region by a junctional zone in mice and rats and the extravillous trophoblast column in humans, respectively [1]. From there, both the interstitial and endovascular trophoblastic cell invasion occurs as gestation advances. In rats and humans, an equally deep invasion takes place in the mesometrial uterus composed of decidua and mesometrial triangle tissue in rats and the inner myometrium in humans, respectively. This region is characterized by a mass of uterine NK cells and numerous loops of spiral arteries [5]. In contrast, a shallower trophoblast invasion into the mesometrial uterus occurs in mice. Hence, these anatomical and physiological differences between mice and humans must be considered for the translation of the results to humans. However, since the first step is to establish the method, the mouse placenta represents a convenient and suitable model system.

### 1.2. Vascular Heterogeneity: Lymphatic Mimicry in the Placenta

In normal tissue, the lymphatic vascularization is critical for a variety of mechanisms, including but not limited to maintaining tissue homeostasis as well as collecting and transferring macromolecules and lymphocytes to the blood circulatory system [10,11]. Although lymphatic and vascular endothelial cells share a common origin as well as endothelial markers such as CD31, the presence of the lymphatic system in the placenta is still under debate [11]. Generally, the co-expression of PROX-1 and CD31 represents a standard analysis to validate the existence of lymphatic endothelial cells. However, no PROX1- and CD31-double-positive cells have yet been identified in the placenta. Nevertheless, Pawlak et al. described how the remodeled spiral artery endothelial cells express the lymphatic markers LYVE1, VEGFR-3, and PROX-1 [12]. The expression of lymphatic markers via spiral artery endothelial cells during the remodeling process (lymphatic mimicry) is considered to play a physiological role during spiral artery remodeling and reduced lymphatic mimicry is associated with poor placentation [12]. To further understand the role of lymphatic vascular heterogeneity and its potential involvement in the pathology of pre-eclampsia, a detailed, subcellular, and spatial visualization of the uteroplacental vasculature, especially of the spiral arteries is needed.

Although adequate placentation is indispensable for a successful pregnancy, the literature is often focused on embryo development [13,14]. Therefore, Elmore et al. provided an extensive histology atlas of the developing mouse placenta [15]. The work was based on two-dimensional sample sections, the current gold standard in histopathology [14]. However, in two-dimensional histopathology, the distribution of the different cell types or the visualization of underrepresented cell types in the tissue sample depends on the slide chosen for analysis. Additionally, the two-dimensional representation of a volumetric architecture is unsuitable for the interpretation of complex spatial structures, connectivity, and network characterization such as the vasculature [16]. Even the three-dimensional reconstruction of subsequently sectioned tissue slides often yields unsatisfactory results along with time-consuming work [10]. Confocal microscopy represents a well-established method for a detailed spatial visualization of thin tissue preparations; however, confocal microscopy is not suited for larger volumetric tissue samples due to the limited axial resolution. Therefore, an optical sectioning approach for the visualization of entire large volumetric samples with an isotropic point-spread-function is required to overcome the aforementioned limitations. Due to orthogonal illumination and detection optics, light sheet microscopy allows for the isotropic illumination and detection of point-spread functions, thus improving the axial resolution of light sheet microscopy compared to confocal microscopy by factor 2–2.5 [15]. Additionally, in light sheet microscopy, image acquisition time is decreased by using CCD cameras instead of point-by-point illumination and detection, as in confocal or multiphoton microscopy. This also reduces the risk of fluorophore bleaching [15]. Therefore, cutting-edge light sheet microscopy meets the necessity for a fast, three-dimensional visualization with a high axial and lateral resolution and minor fluorophore bleaching by optically sectioning entire samples as a z-stack of the tissue sample within seconds. The aim of this work is to establish a robust method for the light sheet microscopy of placental tissue and thus offer a new technique for future projects analyzing the molecular heterogeneity of the uteroplacental vasculature, especially spiral arteries, in pregnancy–disease models (e.g., pre-eclampsia) on a three-dimensional level.

## 2. Materials and Methods

### 2.1. Animals

Laboratory mouse work was approved by the German federal authorities (LaGeSo Berlin, Germany) under the license number ZH120. For this work, C57BL.6/J mice were used. The animals were kept under controlled conditions at a room temperature of 22 °C ± 2 °C, a humidity of 55% ± 10%, and a regulated day/night rhythm (12/12 h). Water and food were available ad libitum.

### 2.2. Preparation of the Placenta

Based on their wide use and availability, C57BL/6J mice were used. Pregnant mice were sacrificed via cervical dislocation at 12.5 days post-coitum (dpc), as the spiral artery remodeling is almost finished at this time point. The placentas were collected and fixated for 4 h in 4% paraformaldehyde (PFA) (*w*/*v* in phosphate-buffered solution (PBS)) at 4 °C. Afterwards, the placentas were manually sectioned at the midline into the slices of 1.5 mm thickness. Slices from the middle third of the placentas were permeabilized (0.5% Triton^TM^ X-100 in PBS (PBS-T)) and blocked in Permablock solution (1% BSA, 0.5% Tween^®^20 in PBS) for two days at 4 °C each.

### 2.3. Whole Mount Immunofluorescence Staining

The immunofluorescence staining of the placenta tissue samples (n (placentas) = 3, n (dams) = 2) were performed at 4 °C under permanent shaking using nanobodies and antibodies. The incubation of primary and secondary antibodies as well as nanobodies and isotype controls was performed in Perm/block solution (eight days each). The LYVE-1 nanobodies were produced in house (Funk et al., unpublished) [16]. After washing three times with PBS-T, the immunostained placenta slices were subjected to dehydration in increasing methanol concentrations (50%, 70%, 95%, >99% (*v*/*v*) methanol in ddH_2_O). Finally, the samples were optically cleared in 1:1 methanol:BABB (benzyl alcohol:benzyl benzoate solution, ratio 1:2) followed by incubation in BABB.

### 2.4. Antibodies

The following antibodies and isolectin were used: rabbit monoclonal anti-His antibody, diluted to 1:200 (12698, Cell Signaling Technologies, Danvers, MA, USA); goat polyclonal anti-mouse/rat CD31 antibody, diluted to 1:200 (AF3628, R&D Systems, Minneapolis, MN, USA); rabbit polyclonal anti-mouse VEGFR-3 antibody, diluted to 1:100 (AF743, R&D Systems, Minneapolis, MN, USA); Syrian hamster monoclonal anti-mouse podoplanin (PDPN) antibody, diluted to 1:200 (clone 8.1.1, Invitrogen, Waltham, MA, USA); rabbit polyclonal anti-human PROX-1 antibody, diluted to 1:200 (RT300-052, ReliaTech GmbH, Wolfenbüttel, DE, Germany); mouse monoclonal anti-human pan-Cytokeratin antibody, diluted to 1:100 (AE1/AE3, Santa Cruz Biotechnology, Dallas, TX, USA), mouse IgG1 kappa Isotype Control, diluted to 1:100 (P3.6.2.8.1, Invitrogen, Waltham, MA, USA) mouse monoclonal anti-human α-smooth muscle actin (α-SMA) antibody, CY3 labeled, diluted to 1:100 (1A4, Sigma-Aldrich, St. Louis, MO, USA); Isolectin (GS-IB4, Invitrogen, Waltham, USA); donkey polyclonal anti-goat IgG Alexa Fluor™ 488 antibody, diluted to 1:1000 (A11055, Invitrogen, Waltham, MA, USA); donkey polyclonal anti-mouse IgG Alexa Fluor™ 488 antibody, diluted to 1:1000 (A21202, Invitrogen, Waltham, MA, USA); donkey polyclonal anti-rabbit IgG Alexa Fluor™ 568 antibody, diluted to 1:1000 (A10042, Invitrogen, Waltham, MA, USA); donkey polyclonal anti-goat IgG Alexa Fluor™ 568 antibody, diluted to 1:1000 (A11057, Invitrogen, Waltham, MA, USA); donkey polyclonal anti-rabbit IgG Alexa Fluor™ 647 antibody, diluted to 1:1000 (A11057, Invitrogen, Waltham, MA, USA); and donkey polyclonal anti-Syrian hamster IgG Alexa Fluor™ 647 antibody, diluted to 1:1000 (A78962, Invitrogen, Waltham, MA, USA).

### 2.5. Whole Mount Fluorescent Hematoxylin and Eosin (H&E) Staining

To perform fluorescent H&E staining, the probes were dehydrated by incubation in a methanol row with increasing concentration (50%, 70%, 95%, and 100% twice) and shaking at room temperature. Subsequently, the fluorescent H&E staining was performed by incubating the samples in acridine orange (AO) and Eosin Y overnight at 4 °C. Finally, the probes were optically cleared as described above. Afterwards, the tissue was stored in ethyl cinnamate at room temperature. Imaging was performed on the same day.

### 2.6. Light Sheet Microscopy

Imaging was performed using a Zeiss Lightsheet 7 (Carl Zeiss Microscopy Deutschland GmbH, Oberkochen, Germany) at various magnifications. The placenta samples were mounted in an upright position onto the sample holder and placed in the corresponding holder in the sample chamber filled with ethyl cinnamate. The subsequent digital three-dimensional reconstruction of the light sheet image stacks was performed using Imaris Viewer (Oxford Instruments plc., Tubney Woods, Abingdon, UK). The qualitative analysis was performed visually using Zen 3.3. blue edition (Carl Zeiss Microscopy Deutschland GmbH, Oberkochen, Germany) and FIJI ImageJ (National Institute of Health, Bethesda, MD, USA). Generally, all samples showed the highest staining intensity at the surface of the sample; however, this can be overcome by computationally adjusting the intensity for every area of interest.

## 3. Results

### 3.1. Whole Mount Fluorescent H&E Staining

In histopathology, the staining of the tissue section using H&E is the current gold-standard method. However, the H&E stain is not suitable for fluorescence-based light sheet microscopy due to insufficient fluorescence spectral properties of the classical H&E dyes. To overcome this, we used AO and Eosin Y as fluorescent H&E analogues dyes. This staining method was previously validated in house with various human tissues in both two-dimensional epifluorescence microscopy and three-dimensional light sheet microscopy (unpublished). For more convenient handling, a computational color remapping was performed, resulting in the well-known violet-pink standard H&E images. The AO and Eosin Y staining showed a complete penetration of the sample, resulting in homogenous staining (Figure 1A(AII)). Thus, the converted fluorescent H&E staining allows the separation between the different functional layers of the placenta, like in conventional H&E staining (Figure 1B(BI)).

### 3.2. Whole Mount Immunofluorescent Staining of Mouse Placental Tissue

Establishing a new method to visualize mouse placental tissue using light sheet microscopy requires an additional evaluation of the imaging results regarding the penetration depth of the fluorophore coupled antibodies as well as the specificity in the context of a volumetric staining [14].

In the scope of this study, placental vascular remodeling, especially lymphatic mimicry, is of high interest. Therefore, eight different target proteins, mostly lymphatic markers, were chosen. The number of stained targets per sample was limited to three by the number of lasers and corresponding detectors of the light sheet microscope. For every staining in this study, a representative image is shown.

To validate the whole mount staining of the 5 × 5 × 1.5 mm placental samples, different immunostaining targeting various markers were applied. We thereby followed a standard protocol to create an overview regarding the feasibility of light sheet microscopy to visualize mouse placental tissue. The following combinations of staining were assessed: PDPN and α-SMA, CD31, isolectin, and PROX-1 and cytokeratin, LYVE-1, and VEGFR-3.

The PDPN and α-SMA double-staining revealed a homogenous staining result with a desirable signal-to-noise ratio across the sample (Figure 2A). Podoplanin, an integral membrane glycoprotein usually expressed in lymphatic endothelial cells, was expressed predominantly in the labyrinthine compartment (Figure 2AII), whereas aside from spiral arteries, only a few PDPN-expressing cells were detected in the mesometrial uterus (Figure 2AI). α-SMA, an actin protein involved in the contractile apparatus of smooth muscles, was also mainly detected in the labyrinth, where PDPN-α-SMA-double-positive cells surrounding the vessel lining were detected (Figure 2BII,CII, respectively, arrowhead). In the mesometrial uterus, only isolated α-SMA-positive and no double-positive cells were detected Figure 2BI).

CD31, isolectin, and PROX-1 triple-staining, on the other hand, revealed a more inhomogeneous staining. While CD31 staining showed a sufficient penetration depth combined with a desirable signal-to-noise ratio, the isolectin penetration remained superficial (Figure 3A,C). Isolectin binds terminal α-galactosyl residues expressed by endothelial cells and is therefore a marker for both lymphatic and blood vessels. Here, a good staining result was obtained at the periphery of the sample, showing the dense vascular network in the labyrinth (Figure 3CII). Besides the labyrinth, endothelial cells in the mesometrial uterus were stained by isolectin (Figure 3CI). Furthermore, the endothelial cells in the mesometrial uterus also expressed CD31, an integral glycoprotein-mediating cell-to-cell adhesion and leukocyte transendothelial migration (Figure 3AI). In the labyrinth, CD31-isolectin double-positive cells were observed (Figure 3DII, arrowheads). Throughout the mesometrial uterus CD31-expressing endothelial cells were also detected, lining the vascular lumen (Figure 3DI, arrowheads). PROX-1, a transcription factor critical for the development and function of the lymphatic system, showed no specific signal, either in the labyrinth or in the mesometrial uterus (Figure 3B–BII).

The placenta sample immunostained for cytokeratin, LYVE-1, and VEGFR-3 showed a high penetration depth and signal-to-noise ratio across the tissue (Figure 4A–C). Cytokeratins are the proteins of cytoskeletal intermediate filaments and were used here as a trophoblast marker. In humans, the fetal trophoblast cells invade the decidua for spiral artery remodeling, partly replacing the endothelial cells lining the lumen [17]. Additionally, the STBs are lining maternal sinusoids in the labyrinth [1]. Here, the pan-cytokeratin staining revealed a strong signal of cells lining the spiral arteries in the mesometrial uterus (Figure 4AI). Furthermore, a dense, vessel-like structure in the labyrinth was detected (Figure 4AII). Respective isotype controls did not result in any staining. A similar result was obtained with the VEGFR-3 staining, which is a cell surface receptor essential for lymph angiogenesis and angiogenic sprouting (Figure 4BI,BII). On the other hand, in the labyrinth, no specific signal was detected for LYVE-1, a ligand-specific hyaluronan receptor on lymphatic endothelial cells (Figure 4CII). Nevertheless, in the mesometrial uterus, LYVE-1 was detected, lining the spiral arteries (Figure 4CI). Therefore, triple-positive cells, such as cytokeratins, LYVE-1, and VEGFR-3, lining the spiral arteries were detected (Figure 4DI, arrowheads). In the labyrinth, only cells double-positive for cytokeratins and VEGFR-3 were detected.

### 3.3. Light Sheet Three-Dimensional Reconstruction

Light sheet microscopy has three major advantages compared to classical histopathology. Firstly, the large number of illuminated planes increases the likelihood of detecting underrepresented cells in a sample. Secondly, the distribution of different cell types can be displayed more robustly, and thirdly, a subsequent three-dimensional reconstruction of the image stack adds a new dimension to analyze a volumetric vascular architecture. Therefore, we performed computational reconstruction for the cytokeratin, LYVE-1, and VEGFR-3 triple-staining and the α-SMA and podoplanin double-staining (Figure 5 and Figure 6). The macroscopic representation reveals a precise visualization of the vascular network. Furthermore, the antibodies were able to penetrate deep into the tissue and showed low unspecific background fluorescence, except from the LYVE-1 staining.

## 4. Discussion

In this study, we were able to establish a new microscopy technique for placental research, especially the spiral arterial remodeling process, which is indispensable for adequate growth of the embryo. By using light sheet microcopy, we were able to add spatial information and thus drastically increase the obtained information per sample. Therefore, this work lays a foundation for three-dimensional studies of spiral artery remodeling. Overall, we were able to confirm the feasibility of light sheet microscopy for murine placental imaging. Furthermore, to validate our novel method, we characterized the expression of major lymphatic markers and were able to show evidence of lymphatic mimicry in murine placentas in three-dimensional samples [18,19].

Here, we were able to establish a reliable approach to apply gold-standard H&E to light sheet microscopy, based on fluorescence. By using AO as a cell nucleus marker, thus replacing hematoxylin, and eosin Y as a counterstain, thus replacing eosin, a specific staining result with high penetration depth and convenient handling was established. Upon a computational color remapping, the well-known pink-violet gold-standard H&E images were generated, thus offering a universally applicable technique to generate three-dimensional H&E images. Hence, this method allows important three-dimensional overviews of the morphology of the placental tissue sample.

### 4.1. Lymphatic Mimicry during Spiral Artery Remodeling Can Be Detected Using Light Sheet Microscopy

First, we investigated whether a lymphatic system is present in murine placentas. Combined staining for CD31 and PROX-1 represents the most reliable way to identify lymphatic endothelial cells [20]. The expression of PROX-1 is mandatory for the maintenance of the lymphatic character of the endothelial cells. Consequently, the overexpression of PROX-1 leads to the transdifferentiation of vascular endothelial cells to lymphatic endothelial cells, whereas the deletion of PROX-1 leads to the transition from lymphatic to vascular endothelial cells, respectively [21,22]. However, in this study we were not able to detect CD31- and PROX-1 double-positive cells in the mouse placenta. Thus, in our study, lymphatic endothelial cells could not be identified in murine placenta.

Second, we investigated whether a lymphatic mimicry could be detected during spiral artery remodeling. We detected VEGFR-3 expression at the innermost layer of the spiral arteries. This has already been described by Pawlak et al. and suggests lymphatic mimicry [12]. VEGFR-3 is the corresponding receptor of angiogenic growth factor VEGF-C, which is predominantly sourced by uNK cells in the decidua during gestation [12]. In fact, VEGFR-3 signaling is required for spiral artery remodeling, underlining the importance of the uNK lymphocytes for placentation [12,23]. In addition, it is known that VEGFR-3 is expressed in humans on invasive cytotrophoblasts in early gestation [24,25]. Here, we were able to confirm that VEGFR-3 is also expressed on the invasive spongiotrophoblasts in the murine decidua. Moreover, previous studies also revealed a decreased VEGFR-3 expression in the mesometrial uterus of sFLT1-mice, a pre-eclampsia mouse model [19]. Consequently, Vogtmann et al. confirmed the reduced lymphatic mimicry upon elevated sFLT1 (soluble fms-like tyrosine kinase-1) levels playing a critical role for the manifestation of typical pre-eclampsia-like maternal symptoms [19]. These findings support the role of lymphatic mimicry in the process of spiral artery remodeling during early pregnancy, as VEGFR-3 in adulthood is predominantly restricted to lymphatic vessels [26]. Additionally, VEGFR-3, as part of the junctional mechanosensory complex, is known to participate in the shear stress-driven remodeling process [27]. Interestingly, VEGFR-3 seems to link an individual shear stress level for different vessel types, known as a shear stress set point, where the endothelial cells respond to a change in the shear by modulating the vessel diameter via remodeling [27]. The shown expression of the VEGFR-3 lining the spiral arteries thereby underlines the role of VEGFR-3 in the remodeling process of spiral arteries and successful placentation, respectively. Furthermore, remodeled spiral arteries and lymphatic vessels also show morphological similarities, including but not limited to the lack of a basement membrane, reduced smooth muscle cell coverage, and a dilated lumen [12]. Consequently, the molecular heterogeneity is critical for the spiral artery to meet the high degree of plasticity during the remodeling process.

Moreover, we were able to show LYVE-1 expression in cells lining the spiral artery lumen, indicating an expression by the murine spongiotrophoblast cells. LYVE-1 is a lymph-specific hyaluronan receptor, playing an important role in regulating cell migration and differentiation during embryogenesis [28,29]. LYVE-1 serves as a lymphatic marker; however, it is a known functional receptor in the placenta, also detected in the lung and liver and on LYVE-1-positive macrophages [30,31]. The functions of both gas exchange and hormone production are similarly performed by the placenta [11]. Hence, our data support the theory that LYVE-1 plays a role beyond hyaluronan transport [11]. In particular, the detected isolated expression of cells lining the spiral arteries supports the work of Pawlak et al., indicating that LYVE-1 might play a key role in spiral artery remodeling [12].

Our data show PDPN expression in proximity to the spiral arteries in the mesometrial uterus. PDPN is a small mucin-like transmembrane glycoprotein that plays a pleiotropic role in inflammation, immune surveillance, epithelial–mesenchymal transition, extracellular matrix remodeling in the periphery, and lymphangiogenesis but not in blood vessel formation [32,33]. It is therefore essential for the formation of lymphatic vessels only. Interestingly, previous studies in humans showed increased VEGFR-3 and reduced PDPN expression in pre-eclamptic placental tissue [34]. The expression of PDPN in cells lining the spiral arteries further indicates a key role of PDPN in spiral artery remodeling. However, the specific role of PDPN in the remodeling process on a molecular level remains unclear. In summary, no lymphatic endothelial cells were identified, but evidence for lymphatic mimicry has been shown. Hence, the spiral arteries are characterized by a molecular heterogeneity during the remodeling process, indicating cellular plasticity, which is important for spiral artery remodeling. In particular, the morphological similarities of lymphatic vessels and remodeled spiral arteries strongly suggest a physiological role for lymphatic mimicry during spiral artery remodeling.

### 4.2. A Spatial Visualization of the Spiral Arteries Is Mandatory for the Study of Fetal Trophoblast Invasion

A functional placenta is essential for the bidirectional exchange of nutrients, gases, and waste products via the intensive intrauterine invasion of trophoblast cells leading to adequate uterine arterial remodeling [1]. The trophoblast cells thereby fulfill three main tasks, including (1) endocrine function to maintain pregnancy, (2) expressing nutrient transporters, and (3) modifying the arterial vasculature [35,36]. Effectively, the trophoblast cells pave the way for the exchange across the maternal–fetal cellular barrier [1]. Hence, a sufficient trophoblast invasion is mandatory for the correct spiral artery remodeling and successful placentation, respectively.

Smooth muscle cells surround the spiral artery endothelial cells in the non-pregnant state; however, in order to increase the uteroplacental blood flow, the smooth muscle cells within the spiral artery walls undergo dedifferentiation. Thereby, the vascular smooth muscle cells swell, disorganize, and are ultimately replaced by fibrinoid and trophoblast cells to increase the lumen. This process is regulated among other factors by the decidua, uterine natural killer cells (uNK), and trophoblast cells [1,35,37].

Hence, for successful placentation, a replacement of the spiral artery smooth muscle cells is mandatory and incomplete dedifferentiation and replacement is associated with pre-eclampsia [13]. Thus, trophoblast cells are an important study target for deeper three-dimensional microscopic analysis.

In our study, the endovascular trophoblast invasion into the decidua was clearly detectable. As expected, endovascular trophoblasts, identified via pan-cytokeratin staining, were lining the spiral arteries, thus replacing endothelial and smooth muscle cells. Here, α-SMA, an actin marker in vascular smooth muscle cells, was used to identify vascular smooth muscle cells [38,39]. Smooth muscle cells were found to surround the fetal sinusoids, whereas in the mesometrial uterus, a dedifferentiation and replacement through trophoblast cells took place. The appearance of intact smooth muscle cell layers in the upper part of the mesometrial uterus underpins the reduced trophoblast invasion in mice compared to other species at the outset. Interstitial trophoblast invasion was also detected; however, the fluorescence intensity was on a lower scale. The spatial visualization of the trophoblast invasion thereby marks a new way of analyzing vascular remodeling by virtually following the vessel through the sample in the future.

### 4.3. Using Endothelial Markers Allows the Three-Dimensional Visualization of the Vascular Architecture

CD31 and isolectin are both markers for endothelial cells. CD31 is a junctional protein with a transmembrane and cytoplasmic domain, expressed on lymphatic and vascular endothelial cells [40,41]. As an endothelial adhesion protein, it regulates endothelial integrity, thereby mediating vascular permeability and lymphangiogenesis as well as angiogenesis [42,43,44]. Due to the high expression of CD31 in endothelial cells, it is a widely used endothelial marker. Thus, a high CD31 expression was expected for the spiral arteries and the fetal blood vessels in the labyrinth, both with a dense endothelial cell lining [14]. By contrast, the maternal sinusoids in the labyrinth are lined with STBs, therefore lacking CD31 expression.

CD31 expression resulted in a homogenous staining, showing the vascular network throughout the labyrinth and decidua. Additionally, isolectin B4 was used to label the alpha-galactose residues of the glycocalyx on the endothelial cells, hence highlighting the matrix surrounding the lymph and blood vessels [45]. However, the superficial staining result of isolectin B4 indicates an insufficient diffusion into the placental tissue sample. To avoid this, an intra-venous injection before sacrificing a mouse could lead to better staining results [46].

A potential limitation of our study is the insufficient penetration depth of antibodies and biomolecules, such as the PROX-1 antibody and isolectin B4, to provide a robust staining result, thus impairing the comparability of these specific staining types. Generally, the placenta is a fairly dense tissue, and therefore, the passive diffusion of the antibodies results in a limited penetration depth. An extended antibody incubation of the immunostaining often results in a higher penetration depth. However, in previous experiments, the chosen tissue size and duration per staining step represents a compromise, as an extended incubation time has not shown any further increase in penetration depth (results not shown). To overcome this limitation for follow-up projects, the usage of fluorophore-coupled single domain antibodies (nanobodies) could provide an alternative, as they are able to diffuse more easily due to their significantly smaller size [16]. The exclusive use of fluorophore-coupled single-domain antibodies has the potential to reduce the inhomogeneity of the staining, enhance the comparability, and finally unleash the power of light sheet microscopy. Furthermore, differences in placental development exist between mouse strains, which must be considered for a translation of the described results to other mouse strains [14,47].

This study analyzed the vascular heterogeneity of the vasculature at a protein level. Combining light sheet microscopy with spatial transcriptomics in future projects could further enhance our knowledge. Additionally, further markers associated with vascularization and lymphatic mimicry could be subject to future analysis [48]. Furthermore, the translation of this methodology to rat and other rodent placentas is the subject of future projects, thus further increasing the translation of the results to human placentas. Moreover, the computational three-dimensional reconstruction holds new opportunities in the near future for an unbiased analysis approach of vascular architectural characteristics based on machine learning.

Finally, by establishing this method, we were able to increase the obtained information per sample, gaining a more complete picture of the fetoplacental physiology. This cutting-edge technology for three-dimensional visualization meets the need for the spatial vascular visualization of the placenta and has the potential to redefine its research. In fact, this work paves the way for consecutive research projects, analyzing the pathology of pre-eclampsia in mouse model systems to reveal new insights into the protein expression in pre-eclamptic placentas, find new potential targets, and ultimately evaluate treatment options.

## Figures and Tables

**Figure 1 biomolecules-13-01009-f001:**
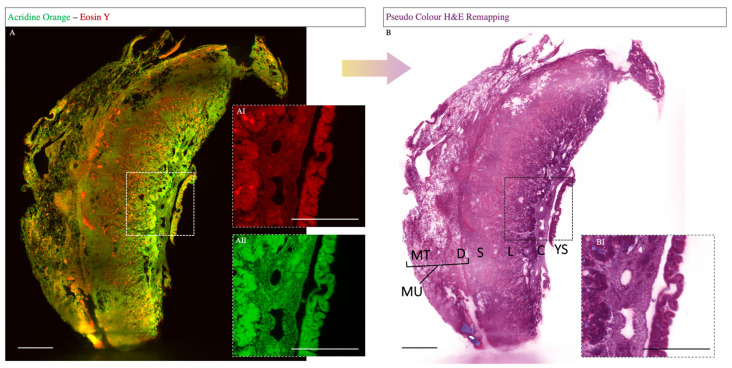
**Whole mount fluorescent H&E staining of placental tissue.** The pseudo-color H&E image stack shows the expected histological compartments of the definitive placenta (12.5 dpc): YS—yolk sac; C—chorionic plate; L—labyrinth; S—spongiotrophoblast; D—decidua; MT—mesometrial triangle tissue; MU—mesometrial uterus. Fluorescent H&E staining (**A**) with acridine orange (**AI**) and eosin Y (**AII**) used as raw data for the computational color remapping (**B**). The converted H&E staining shows a specific staining of the violet nuclei and a pink counterstaining (**BI**). Scale bars: 500 µm.

**Figure 2 biomolecules-13-01009-f002:**
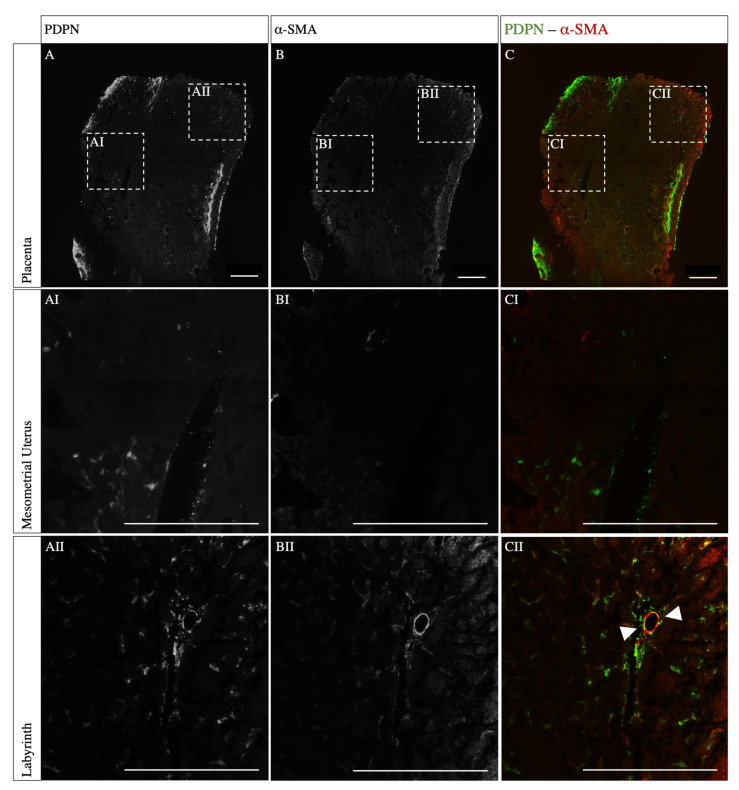
**Co-staining of whole mount placental tissue for PDPN and α-SMA**. The PDPN-α-SMA double-staining shows a homogenous result throughout the sample. Mouse placenta (12.5 dpc) was immunostained for PDPN (**A**), α-SMA (**B**), and merged overlay (**C**). (**AI**–**CI**) show α-SMA-positive cells around the spiral arteries and PDPN-positive cells in proximity to the spiral arteries in the mesometrial uterus with no double-positive cells. (**AII**–**CII**) show PDPN-positive cells surrounding α-SMA expressing vascular endothelial cells in the labyrinth, and double-positive cells are marked with an arrowhead. Scale bars: 500 µm.

**Figure 3 biomolecules-13-01009-f003:**
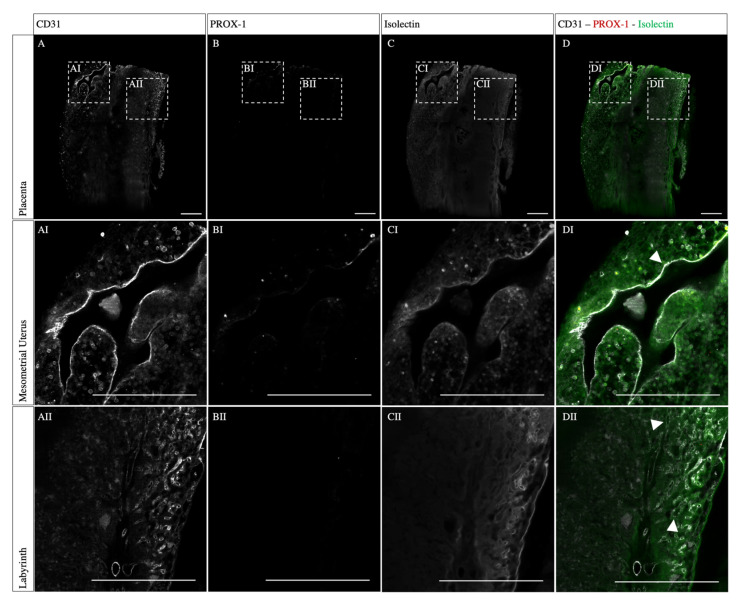
**Co-staining of whole mount placental tissue for CD31, PROX-1, and isolectin.** The staining for CD31 revealed a homogenous staining result covering the complete sample, whereas the isolectin staining stayed superficial and no PROX1 expression could be detected. Mouse placenta (12.5 dpc) immunostained for CD31 (**A**), PROX-1 (**B**), Isolectin (**C**), and merged overlay (**D**). (**AI**–**DI**) shows CD31 and double-positive isolectin cells lining a vascular lumen, exemplarily marked with an arrowhead. (**AII**–**DII**) show double-positive cells lining the dense vascular structures in the labyrinth, exemplarily marked with a arrowheads. Scale bars: 500 µm.

**Figure 4 biomolecules-13-01009-f004:**
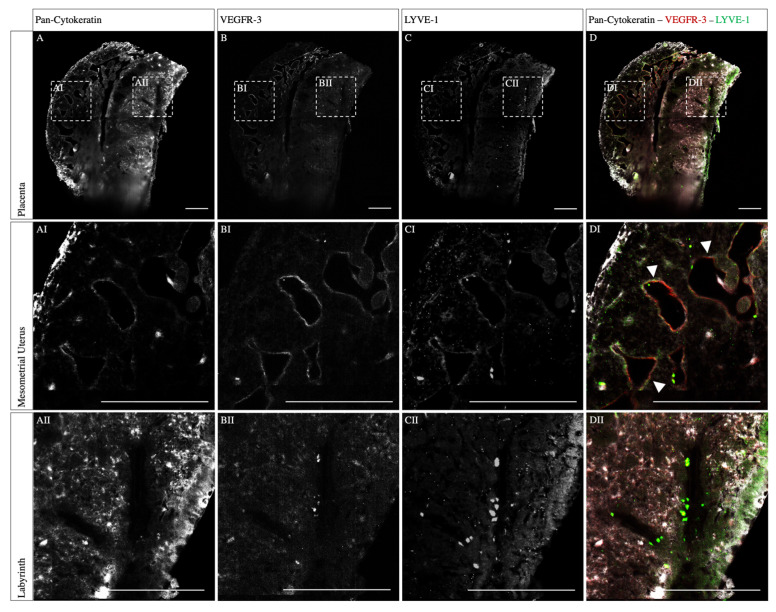
**Co-staining of whole mount placental tissue for pan-cytokeratin, VEGFR-3, and LYVE-1.** The staining for pan-cytokeratin, LYVE-1, and VEGFR-3 revealed a complete and homogenous staining result. Mouse placenta (12.5 dpc) immunostained for cytokeratin (**A**), VEGFR-3 (**B**), LYVE-1 (**C**), and merged overlay (**D**). (**AI**–**DI**) show unspecific cytokeratin staining in the periphery of the sample and triple-positive cells lining the lumen of spiral arteries (arrowheads) in the mesometrial uterus. (**AII**–**DII**) show cells expressing cytokeratin and VEGFR-3 but no LYVE-1 expression in the labyrinth. Scale bars: 500 µm.

**Figure 5 biomolecules-13-01009-f005:**
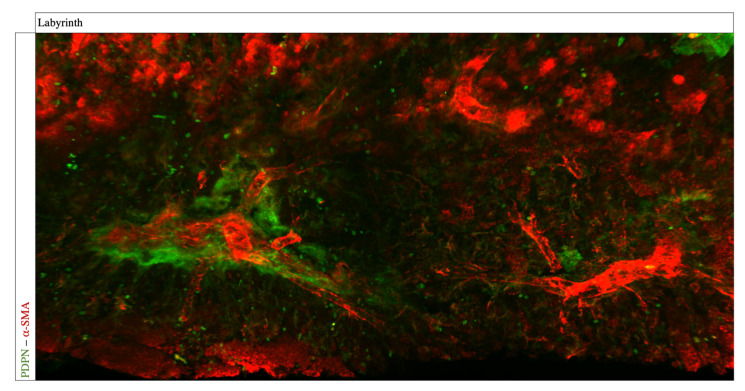
**Three-dimensional reconstruction of whole mount placental tissue for PDPN and α-SMA.** By converting the two-dimensional images into a three-dimensional volume, spatial classification is possible. The whole mount immunofluorescence staining of a mouse placenta (12.5 dpc) for α-SMA (red) and podoplanin (green). Image was taken using light sheet microscopy. Shown is a computational volumetric reconstruction, revealing the three-dimensional architecture of vessels in the labyrinth with the podoplanin-expressing cells surrounding the vessels.

**Figure 6 biomolecules-13-01009-f006:**
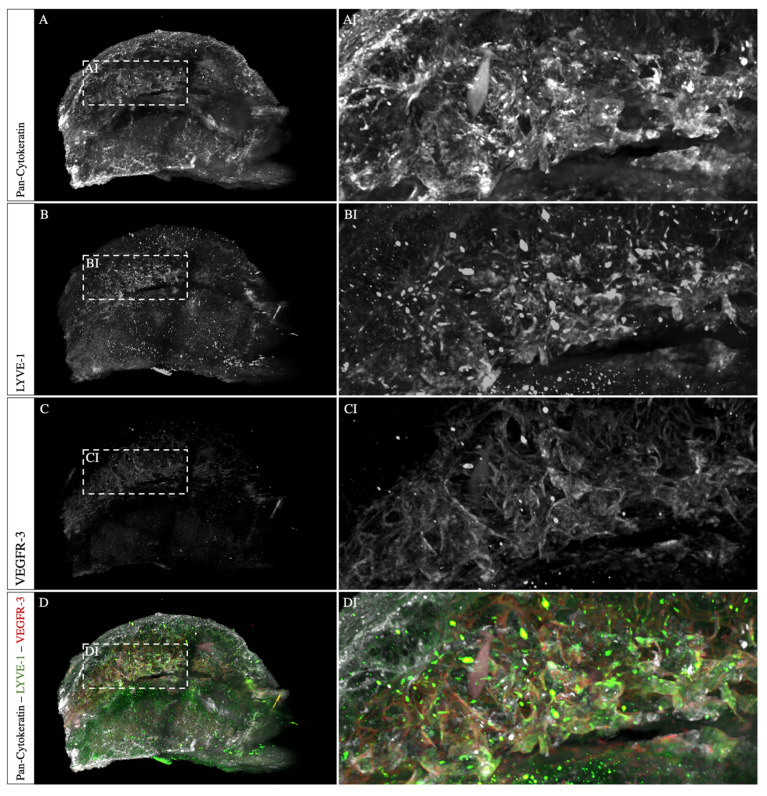
**Light sheet images allow a three-dimensional visualization of the stained sample.** The whole mount immunofluorescence staining of a mouse placenta (12.5 dpc) for cytokeratin (**A**), LYVE-1 (**B**), VEGFR-3 (**C**), and merged (**D**). Shown are computational volumetric reconstructions, revealing the three-dimensional architecture of the vessels. The zoom-in shows the previously described triple-positive vascular structure (**AI**–**DI**).

## Data Availability

The data that support the findings of this study are available from the corresponding authors upon reasonable request.

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
