# Peer review of "Three-Dimensional Histological Characterization of the Placental Vasculature Using Light Sheet Microscopy"

_biomolecules, 2023, doi:10.3390/biom13061009_

Round 1

Reviewer 1 Report

This paper aims to use light sheet microscopy to study the process of reconstitution of spiral artery in mouse placenta, but the results seem to be insufficient. The reasons may be technical difficulties such as antibody permeability, but I think it is also due to inadequate experimental materials and preparation, misunderstanding of basic knowledge, and insufficient setting of experimental objectives, as described below. Of course, this may not be the case, but at least that is what I have to conclude from your paper.

Major comments

1. The photo of placentas in this paper did not include a clear tissue of the metrial gland. Did you confirm the presence of a metrial gland in the sections by identifying uNK cells? Placenta should be sectioned at the midline to observe the metrial gland (MG) and spiral artery (SA). For example, the AI area in Figure 2 is the region of decidua basalis and characteristic dilated veins, not the MG. The SA are not present at the peripheral area of the placenta. I predict that the placental sections used in this study were not suitable for analyses.

2. I am confident that there is no distribution of SA or no infiltration of trophoblast into the area of MG.  The MG corresponds to the circular muscle layer at the non-pregnant periods, and if trophoblast invade into this area, it would be considered an abnormal infiltration, such as that of cancer cells. Without seeing any supporting papers or data, it is difficult for me to agree the authors' knowledge on which this paper is based.

3. In the introduction, the authors state to attempt a "volumetric analysis" for blood vessels using light sheet microscopy, but they do not provide specific data for quantitative analysis, so the novelty of this paper is not clear.

4. The authors focused on "lymphatic mimicry", but authors did not explain what this phenomenon is, and it is difficult to understand the significance of their choice of experimental technique based on "lymphatic mimicry. The authors' intention was also unclear as to whether they wanted to see at blood vessels or lymphatic vessels.

5. In Figures 1-4, it is difficult to determine whether there is a positive reaction to vascular vessels at the area other than the spiral artery. It is possible that the magnification of the photograph is too small.

6. The authors refer to the staining result of the spiral artery as "molecular heterogeneity" (L363-364, L388), and also refer to it as "double positive" in L262. On the other hand, it was referred as an "artifact (L456-458). The authors' views are not consistent.

Minor comments

L109: I don't know what "visualization of rare cells" means.

L124: "C57B1.6/J" --> C57BL/6J

L129, L131: "C57BL/6" --> C57BL/6J or abbreviation as B6

L130: "differences of placentation" needs explanation.

L157 "donkey polyclonal anti-mouse IgG Alexa Fluor™ 488 antibody":  Authors did not describe how to avoid auto-reaction on the mouse tissues.

Figure 1B, 1C: BI and BII are incorrectly positioned in figure 2B, they are interchanged. Same for CI and CII in figure 2C.

L290: cores ---> nuclei

L311: No explanation for "Sb".

L403-410: Author should explain what kind of change is "dedifferentiation of spiral artery"?

Abbreviation: First time should indicate the full spelling in PDPN, SMA, etc.

Author Response

We thank the reviewer 1 for taking the time to review our manuscript and appreciate your criticism and valuable feedback. We edited the manuscript upon your valuable suggestions and hope that it is now suitable for publication.

Reviewer 2 Report

In the manuscript "Three-Dimensional Histological Characterization of the Placental Vasculature Using Light Sheet Microscopy" Freise et al. set out to employ Light Sheet Microscopy to examine aspects of placental development and compare their observations to those derived using standard techniques. This exciting and pioneering study applies a new technique to examine critical aspects of placentation that may help researchers better understand fundamental biological processes involved in placental development and function. However, this paper needs to be better organized, tell a focused story, include critical methodological details expected of a technical paper, and present convincing evidence that this technique works. In its current form, this study is unsuitable for publication.

First and foremost, this is a technical paper. Therefore, the introduction needs to discuss confocal microscopy, highlight the relevance to studies of placental biology, emphasize the technical limitations of this technique, then introduce light sheet microscopy, explain this new technique, and contrast this form of microscopy with confocal so that readers can understand why the authors are conducting this work. The authors need to significantly abbreviate their discussions of placental biology and disease, then tell readers the story of light sheet vs. confocal microscopy.  

This manuscript describes a novel approach to analyzing mouse placental development and function. Therefore, the authors need to measure established, well-characterized features of the mouse placenta and compare their resolution, reproducibility, and quantitative measures between confocal and light sheet techniques. Why did the authors choose this study to chase questions concerning the presence of the lymph system when the presence or function of this system in the placenta is debated? When validating a new protocol, readers want to compare established measures of histological organization, antibody staining, and architecture between the current (confocal) and new (light sheet) techniques. The authors must employ established antibodies and staining techniques and validate their staining using established 2D IHC. Otherwise, readers lack convincing proof that the author's report is accurate. This reviewer has enthusiasm for this work - but when developing a new technique, go slow and use established measures to prove the validity and reproducibility of your observations.    

Why did the authors cut the placentas into 1.5mm slices? The stated benefit of light sheet microscopy and the core premise of the paper is the need for detailed, non-destructive visualization of the placenta. There are several published examples of entire adult mouse brain and body clearing; why not here?

Why did the authors not use DAPI staining to confirm antibody penetrance?

The authors must include representative images to prove they achieved complete tissue clearing.

The authors present multiple subjective observations as proof of antibody success and tissue penetrance. However, readers need to see analysis using quantitative methods.

Overall, the protocol could have been more specific and detailed. For example, what was the duration of immunostaining? How many placentas did the authors use? In addition, there needed to be more explanation for why the authors chose specific methods or solutions over other alternatives.

Minor Points:
Figure 1. A1-C1 and A2-C2 should be switched in figure legend or Figure 1: C1 and C2 swapped.

Author Response

We thank the reviewer 2 for taking the time to review our manuscript and appreciate your criticism and valuable feedback. We edited the manuscript upon your valuable suggestions and hope that it is now suitable for publication.

Reviewer 3 Report

Only a few very minor remarks:

Abstract

31 did you mean “representing” instead of “resembling”

Introduction

51 did you mean “represents” instead of “resembles”

56 “Stillbirth” may be better than miscarriage because placental etiologies usually impact pregnancies later (the most common etiology of miscarriage in humans is sporadic aneuploidy)

6o maybe specify “mouse placenta”

87  did you mean “represents” instead of “resembles”

386 Maybe change to:  “In summary, no lymphatic endothelial cells were identified, but lymphatic mimicry could be clearly proven:.

399 ,maybe change to: “pave the way”

Meaning and content:

Abstract

34- please specify if the 16X increase in blood flow is in rodents/ animal models or in human

Figure 1

I think several of the individual images are mislabeled in this composite figure. Please review

Author Response

We thank the reviewer 3 for taking the time to review our manuscript and appreciate your criticism and valuable feedback. We edited the manuscript upon your valuable suggestions and hope that it is now suitable for publication.
